# Transcription activation by a sliding clamp

Jing Shi[1,2,4], Aijia Wen[1,4], Sha Jin[1,4], Bo Gao[1], Yang Huang[1] & Yu Feng [1,3✉]

Transcription activation of bacteriophage T4 late genes is accomplished by a transcription activation complex containing RNA polymerase (RNAP), the promoter specificity factor gp55, the coactivator gp33, and a universal component of cellular DNA replication, the sliding clamp gp45. Although genetic and biochemical studies have elucidated many aspects of T4 late gene transcription, no precise structure of the transcription machinery in the process is available. Here, we report the cryo-EM structures of a gp55-dependent RNAP-promoter open complex and an intact gp45-dependent transcription activation complex. The structures reveal the interactions between gp55 and the promoter DNA that mediate the recognition of T4 late promoters. In addition to the σR2 homology domain, gp55 has a helix-loop-helix motif that chaperons the template-strand single-stranded DNA of the transcription bubble. Gp33 contacts both RNAP and the upstream double-stranded DNA. Gp45 encircles the DNA and tethers RNAP to it, supporting the idea that gp45 switches the promoter search from three-dimensional diffusion mode to one-dimensional scanning mode.

[1] Department of Biophysics, and Department of Pathology of Sir Run Run Shaw Hospital, Zhejiang University School of Medicine, Hangzhou, China.
[2] Department of Microbiology and Immunology, School of Medicine & Holistic Integrative Medicine, Nanjing University of Chinese Medicine, Nanjing, China.
[3] Zhejiang Provincial Key Laboratory of Immunity and Inflammatory diseases, Hangzhou, China. [4] These authors contributed equally: Jing Shi, Aijia Wen, Sha Jin. ✉email: yufengjay@zju.edu.cn

To initiate the transcription of most genes, *E. coli* RNA polymerase (RNAP, subunit composition $\alpha_2\beta\beta'\omega$) forms holoenzyme with $\sigma^{70}$, which contacts RNAP extensively and recognizes the promoter DNA[1–7]. Specifically, $\sigma$ conserved region $\sigma$R2 contacts the clamp helices of the RNAP $\beta'$ subunit and recognizes the promoter -10 element, while $\sigma$ conserved region $\sigma$R4 contacts the flap tip helix (FTH) of the RNAP $\beta$ subunit and recognizes the promoter -35 element.

Bacteriophages use bacterial RNAP to transcribe their own genes. For decades, transcription of bacteriophage T4 late genes has served as a model system to investigate mechanisms of transcription regulation. The T4 late promoters consist of a -10-like element placed at the position corresponding to the bacterial promoter -10 element[8]. Nevertheless, there is no sequence conservation at the position corresponding to the bacterial promoter -35 element. Herendeen et al. developed an in vitro system that leads to the current understanding of T4 late transcription[9]. It was found that T4 encoded gp55 enabled RNAP to execute low level (basal) transcription, which was further activated by the sliding clamp gp45 and its coactivator gp33.

Gp55, a highly diverged $\sigma^{70}$ family protein[10], plays an essential role in T4 late transcription. To initiate basal transcription, bacterial RNAP needs to form holoenzyme with gp55, which recognizes T4 late promoters and confers the ability to form a catalytically competent RNAP-promoter open complex (RPo-gp55)[11]. The RNAP-binding motif of gp55 has been inferred on the basis of alanine scanning mutagenesis analyzed for RNAP binding, basal and activated transcription[12]. Initial binding of RNAP–gp55 holoenzyme to DNA is not sequence-specific, while RPo-gp55 is sequence-specific and readily detected by footprinting[13,14]. The acquisition of sequence specificity on promoter opening implies recognition of some feature of the transcription bubble by gp55, but this has not been demonstrated directly.

Gp33 and gp45 further activate the basal transcription activity of RNAP–gp55 holoenzyme by forming a catalytically competent transcription activation complex (TAC-gp45)[14]. Pull-down experiments showed that gp33 binds to the RNAP $\beta$ flap[15], which was further confirmed by the crystal structure of gp33 complexed with the RNAP $\beta$ flap[16]. Although gp33 does not, by itself, bind to DNA[9], photo-crosslinking experiments showed that gp33 was proximal to the upstream double-stranded DNA (dsDNA) in TAC-gp45[14,17]. Without a precise structure of TAC-gp45, whether gp33 directly contacts the upstream dsDNA is unknown. DNase I footprinting and photo-crosslinking experiments collectively suggested that gp45 was located at the upstream end of TAC-gp45, in the vicinity of the coactivator, gp33[14]. Gp45 is a homotrimer, forming a triangle with a central hole large enough to accommodate dsDNA[18]. The lateral face with the protruding C-terminus presents a hydrophobic patch on each protomer that serves as a binding site for the sliding clamp-binding motif (SCBM) that is attached to the body of gp55 and gp33 through a linker. In vitro transcription assays with SCBM truncated proteins showed that the SCBM of gp33 is essential for transcription activation, while loss of the SCBM of gp55 impairs but does not abolish transcription activation[19,20].

Although genetic and biochemical studies have elucidated many aspects of T4 late gene transcription, no precise structure of the transcription machinery in the process is available. Here, we present the cryo-EM structure of RPo-gp55, revealing the structural basis for the basal transcription activity of RNAP–gp55 holoenzyme. We also determine the cryo-EM structure of an intact TAC-gp45 that provides a structural framework for interpreting previous results and serves as a guide for the design of further experiments.

## Results

### Cryo-EM structures of RPo-gp55 and TAC-gp45.
*E. coli* RNAP, gp55, gp33, gp45, clamp loader gp44/gp62, and single-stranded DNA (ssDNA)-binding protein gp32 are purified to homogeneity (Supplementary Fig. 1a). To ascertain the activity of the purified proteins, we develop an in vitro transcription assay by taking advantage of a fluorogenic RNA aptamer, Mango III (Supplementary Fig. 1b)[21]. The nucleic acid scaffold encompasses the essential features of a T4 late transcription unit that can be activated by gp45 (Fig. 1a). It includes the T4 late promoter $P_{23}$ followed by Mango III encoding sequence and *rrnB* terminators (*T1* and *T2*). Loading of gp45 onto this DNA with the appropriate polarity for transcription activation is assured by ~150 nt of 5′ overhanging ssDNA, generated by exonuclease III, at the downstream DNA end only. The transcription activities of RNAP in the presence of gp55, gp33, and gp45 are evaluated by measuring the fluorescence intensities of Mango III (Supplementary Fig. 1c). In accordance with previous studies[13,22], RNAP–gp55 holoenzyme initiates basal transcription. Gp33 alone represses gp55-dependent transcription, while the combination of gp33 and gp45 activates gp55-dependent transcription. The level of enhancement observed in Mango III experiments (~3-fold) is similar to the previous work using radioactive transcription assay[13,16,22].

To obtain a structure of the T4 late gene transcription machinery, we incubate *E. coli* RNAP, gp55, gp33, gp45, gp44/gp62, gp32, and the nucleic acid scaffold, freeze the sample, and collect data on Titan Krios. Steps of 3D classification reveal the structures of RPo-gp55 and TAC-gp45 (Fig. 1, Supplementary Figs. 2–5, and Supplementary Table 1). The structure of RPo-gp55 is determined at a nominal resolution of 3.6 Å (Supplementary Fig. 3a). Local resolution calculation indicates that the central core of the structure is determined to 3.0–4.0 Å resolution (Supplementary Fig. 3c). The structure of TAC-gp45 is determined at a nominal resolution of 4.5 Å (Supplementary Fig. 4a). Local resolution calculation indicates that the central core of the structure is determined to be 4.0–6.0 Å (Supplementary Fig. 4c).

Experimental density maps show clear densities for RNAP, gp55, gp33, gp45, and the nucleic acid scaffold (Fig. 1b, c and Supplementary Fig. 5). The RNAPs of the structures are very similar to the previously reported $\sigma^{70}$-dependent RNAP-promoter open complex (RPo-$\sigma^{70}$) structure[6]. Consistent with the previous KMnO$_4$ footprinting experiments[13], nucleic acid scaffold from −12 to +2 is unwound in RPo-gp55 and TAC-gp45, while the upstream dsDNA and the downstream dsDNA adopt similar orientations as in RPo-$\sigma^{70}$. Gp55 adopts the same conformation and makes essentially identical interactions with RNAP and the promoter DNA in RPo-gp55 and TAC-gp45. The interactions in RPo-gp55 will be discussed in the following sections due to its superior resolution.

### σR2 homology domain of gp55 contacts the clamp helices.
Gp55 is composed of five helices (H1–H5) connected by short linkers (Fig. 2a). As predicted by the sequence analysis[10], the folding of H1–H4 is reminiscent of $\sigma$R2 (Fig. 2b, c and Supplementary Fig. 5a). However, H4 is much longer than the equivalent helix in $\sigma$R2 and forms a helix-loop-helix (HLH) motif, together with H5 and their linker.

There is no clear density for two terminal segments of gp55. In an alignment of gp55 homologs, the N-terminal region is poorly conserved, and many homologs lack this region entirely[23]. The C-terminal region of gp55 contains the SCBM, which would not be expected to interact with RNAP or DNA, but instead to be available for interaction with gp45. Accordingly, the C-terminal region truncated gp55 binds RNAP, forms RPo-gp55, and

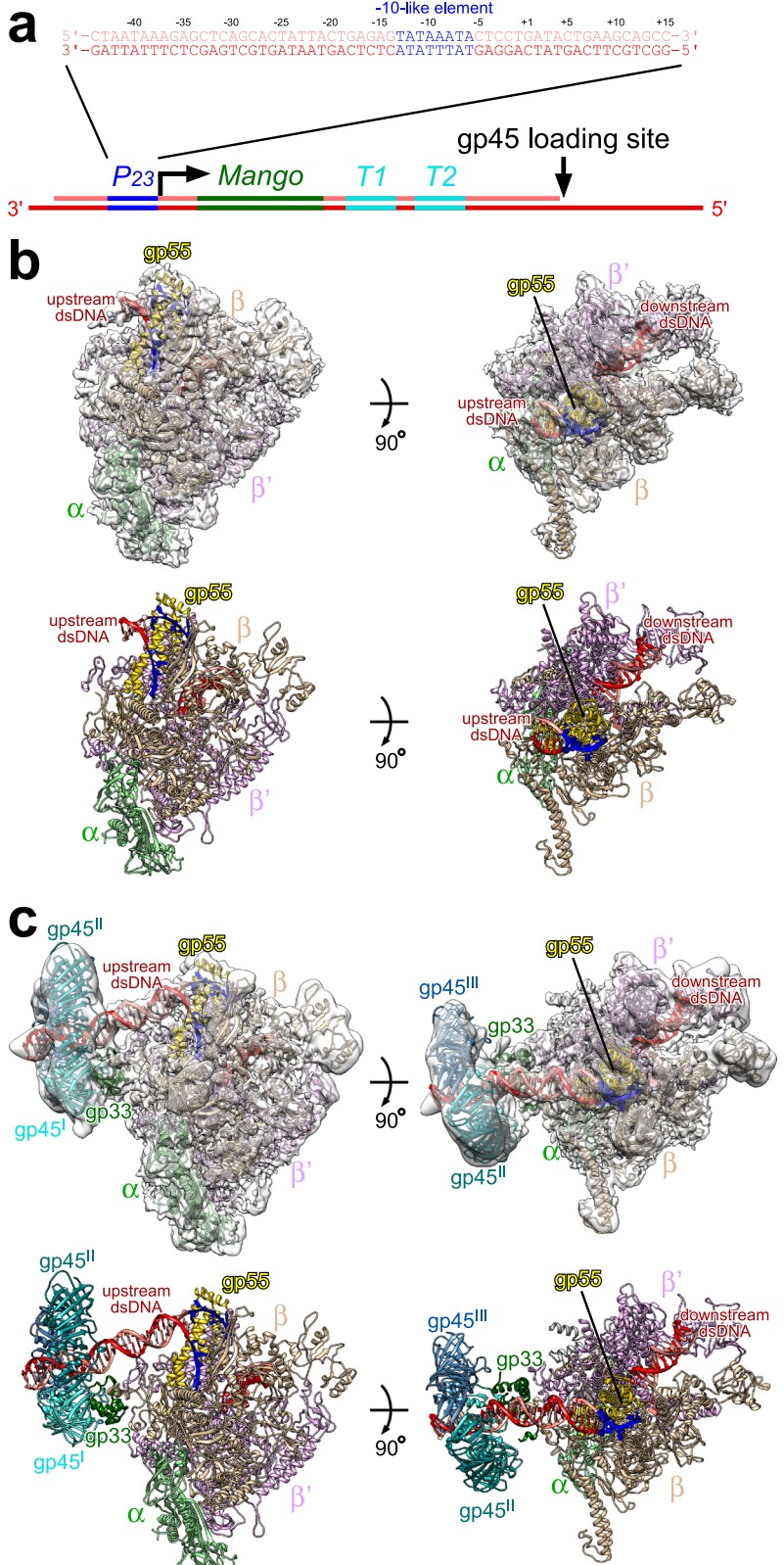

**Fig. 1 Cryo-EM structures of RPo-gp55 and TAC-gp45. a** Nucleic acid scaffold used for cryo-EM. The nucleic acid scaffold encompasses the T4 late promoter $P_{23}$ followed by Mango III encoding sequence and *rrnB* terminators (*T1* and *T2*). Loading of gp45 onto this DNA with the appropriate polarity for transcription activation is assured by ~150 nt of 5′ overhanging ssDNA at the downstream DNA end only. Salmon, nontemplate-strand DNA; red, template-strand DNA; blue, $P_{23}$; green, *Mango*; cyan; *T1* and *T2*. Positions are numbered relative to the transcription start site. **b** The cryo-EM density map and the superimposed model of RPo-gp55. Light green, RNAP α subunits; wheat, RNAP β subunit; light pink, RNAP β′ subunit; yellow, gp55; salmon, nontemplate-strand DNA; red, template-strand DNA; blue, -10-like element; dsDNA, double-stranded DNA. **c** The cryo-EM density map and the superimposed model of TAC-gp45. Dark green, gp33; cyan, gp45[I]; turquoise, gp45[II]; blue, gp45[III]. View orientation and other colors as in **b**.

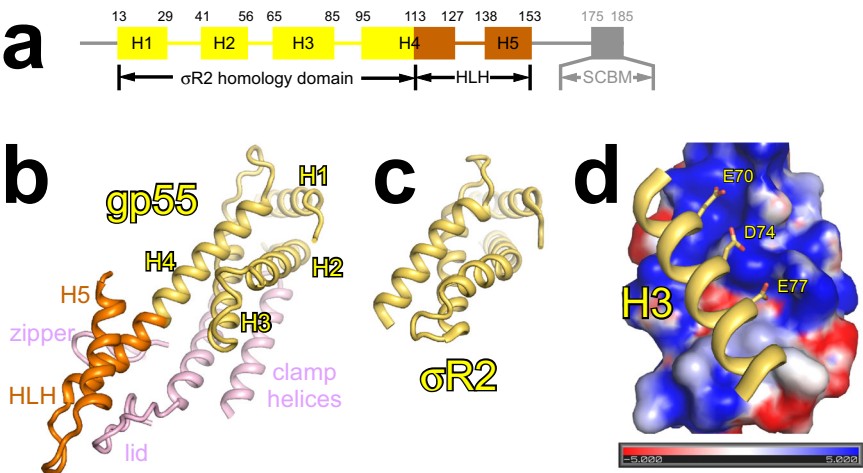

**Fig. 2 The σR2 homology domain of gp55 contacts the clamp helices. a** Structural organization of gp55. Two terminal segments are disordered and colored gray. Observed secondary structure features (H1–H5) are shown schematically and labeled. Yellow, σR2 homology domain; orange, helix-loop-helix (HLH) motif; SCBM, sliding clamp-binding motif. **b** The σR2 homology domain contacts the clamp helices. Yellow, σR2 homology domain; orange, helix-loop-helix (HLH) motif; light pink, clamp helices, zipper, and lid. **c** σR2 in RPo-σ70 (PDB 6CA0 [https://doi.org/10.2210/pdb6CA0/pdb]). View orientation as in **b**. **d** Three negatively charged residues of H3 make electrostatic interactions with a positively charged groove of the clamp helices. Blue represents positively charged surfaces (+5 kT) and red represents negatively charged surfaces (−5 kT).

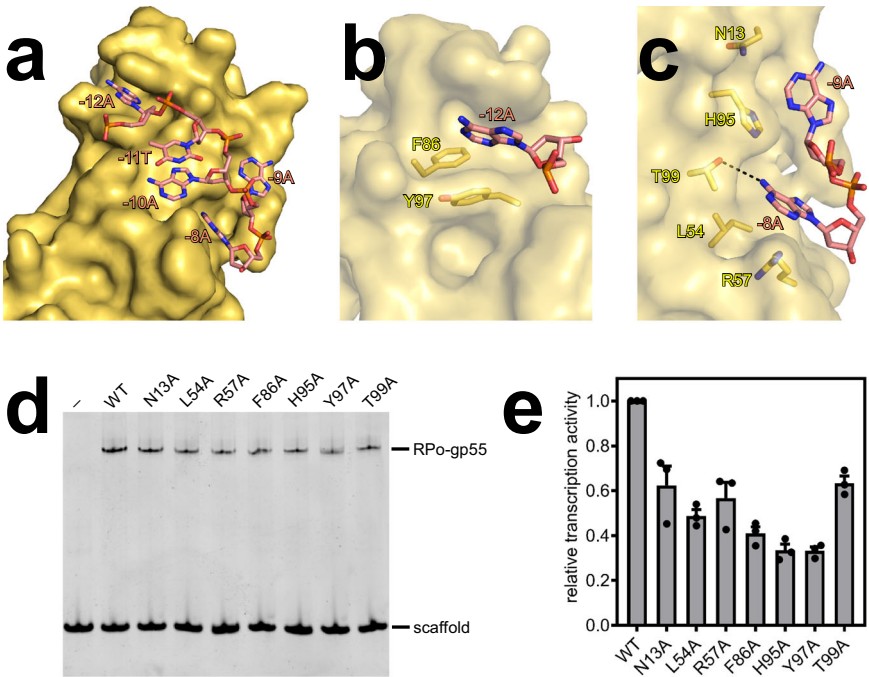

**Fig. 3 The σR2 homology domain of gp55 mediates the recognition of the promoter -10-like element. a** Interactions of gp55 with nontemplate-strand ssDNA. Gp55 is shown as surfaces and DNA is shown as sticks. **b** For promoter position -12, the base is unstacked and inserted into a protein pocket. **c** For promoter positions -9 and -8, the bases are unstacked and inserted into protein pockets. **d** Electrophoretic mobility shift assays show that substitutions of the pocket residues impair RPo-gp55 formation. Experiments were repeated independently three times with similar results. **e** Mango III transcription assays show that substitutions of the pocket residues jeopardize gp55-dependent transcription. Error bars represent mean ± SEM out of $n = 3$ experiments.

initiates basal transcription as effectively as full-length gp55 (Supplementary Fig. 6).

The σR2 homology domain of gp55 binds to the clamp helices of RNAP. In particular, three negatively charged residues (E70, D74, E77) of H3 make electrostatic interactions with a positively charged groove of the clamp helices (Fig. 2d and Supplementary Fig. 5b), consistent with the previous report that amino acid substitutions of these residues strongly disrupt the gp55–RNAP interaction in competitive binding experiments[12].

**σR2 homology domain of gp55 mediates the recognition of the promoter -10-like element**. The σR2 homology domain of gp55 mediates the recognition of the promoter -10-like element through interactions with nontemplate-strand ssDNA in the unwound transcription bubble (Fig. 3a and Supplementary Fig. 5c). The σR2 homology domain of gp55 unstacks nucleotides, flips nucleotides, and inserts nucleotides into protein pockets at positions -12, -9, and -8. In particular, the adenine at position -12 is inserted into a deep protein pocket, making a stacking

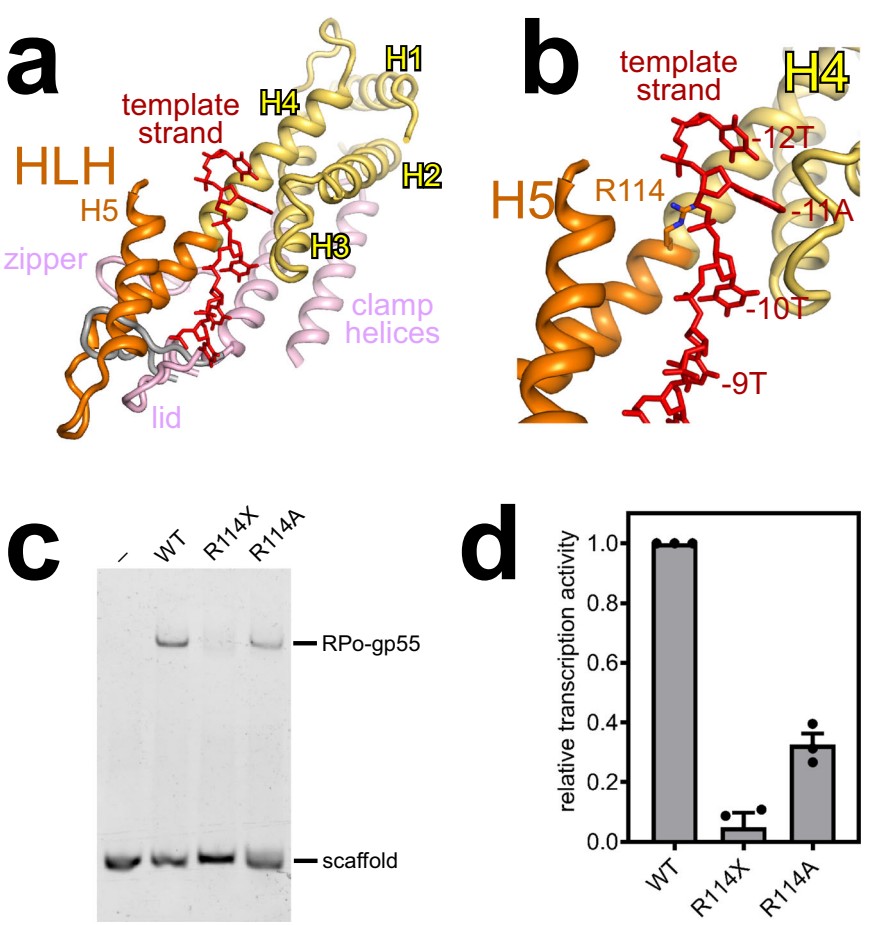

**Fig. 4 The helix-loop-helix (HLH) motif of gp55 chaperons the template-strand ssDNA of the transcription bubble. a** The HLH motif of gp55 contacts the zipper, displaces the lid of RNAP, and chaperons the template-strand ssDNA of the transcription bubble. Gray, lid in RPo-σ[70] (PDB 6CA0 [https://doi.org/10.2210/pdb6CA0/pdb]); red, template-strand ssDNA. View orientation and other colors as in Fig. 2b. **b** Gp55 residue R114 makes an electrostatic interaction with the DNA backbone phosphate at position -11. Colors as in **a**. **c** Electrophoretic mobility shift assays show that both truncation of the HLH motif and substitution of residue R114 impair RPo-gp55 formation. Experiments were repeated independently three times with similar results. **d** Mango III transcription assays show that both truncation of the HLH motif and substitution of residue R114 jeopardize gp55-dependent transcription. Error bars represent mean ± SEM out of $n = 3$ experiments.

interaction with residue Y97 (Fig. 3b). The adenine at position -9 is inserted into a shallow protein pocket, making van der Waals interactions with residues N13 and H95, while the adenine at position -8 is inserted into a deep protein pocket, making a H-bonded interaction with residue T99 (Fig. 3c). Alanine substitutions of residues (N13, L54, R57, F86, H95, Y97, and T99) in these protein pockets impair RPo-gp55 formation and gp55-dependent transcription (Fig. 3d, e), verifying that the cryo-EM structures are biologically relevant.

**HLH motif of gp55 chaperons the template-strand ssDNA of the transcription bubble.** The HLH motif of gp55 contacts the zipper and displaces the lid of RNAP (Fig. 4a). Although truncation of the HLH motif does not affect RNAP–gp55 binding in pull-down experiments (Supplementary Fig. 6a, b), the HLH motif may contribute to RNAP–gp55 holoenzyme formation in vivo.

Strikingly, the HLH motif chaperons the template-strand ssDNA of the transcription bubble (Fig. 4a and Supplementary Fig. 5d). In particular, residue R114 makes an electrostatic interaction with the DNA backbone phosphate at position -11 (Fig. 4b). Therefore, we infer that the HLH motif probably contributes to T4 late transcription by stabilizing the template-

strand ssDNA of the transcription bubble. Consistently, both truncation of the HLH motif and substitution of residue R114 compromise RPo-gp55 formation and gp55-dependent transcription (Fig. 4c, d).

**Interactions between gp33 and RNAP.** Gp33 is composed of five helices (H1–H5) connected by short loops, as well (Fig. 5a and Supplementary Fig. 5e). Early work indicated that gp33 bound to the RNAP β flap and a crystal structure of gp33 in complex with the RNAP β flap has been determined[15,16]. The structure of gp33 and β flap in TAC-gp45 is superimposable on the crystal structure (neglecting βi9, which does not contribute to gp33–β flap interaction; Fig. 5a), indicating that gp33 makes a similar set of interactions in both structures. Although gp33 and σR4 are not structurally related, gp33 clamps onto the FTH as σR4. Substitutions of some of the interacting residues (gp33 residue E70; FTH residues K900, L901, L902, I905, and F906) have been demonstrated to impair gp33–β flap interaction in a bacterial two-hybrid assay[16].

When bound to the β flap, the negatively charged residues E64 and E65 of gp33 are positioned to make electrostatic interactions with the positively charged residues K76, R77, and K79 of the RNAP zinc-binding domain (Fig. 5b). In accordance, alanine

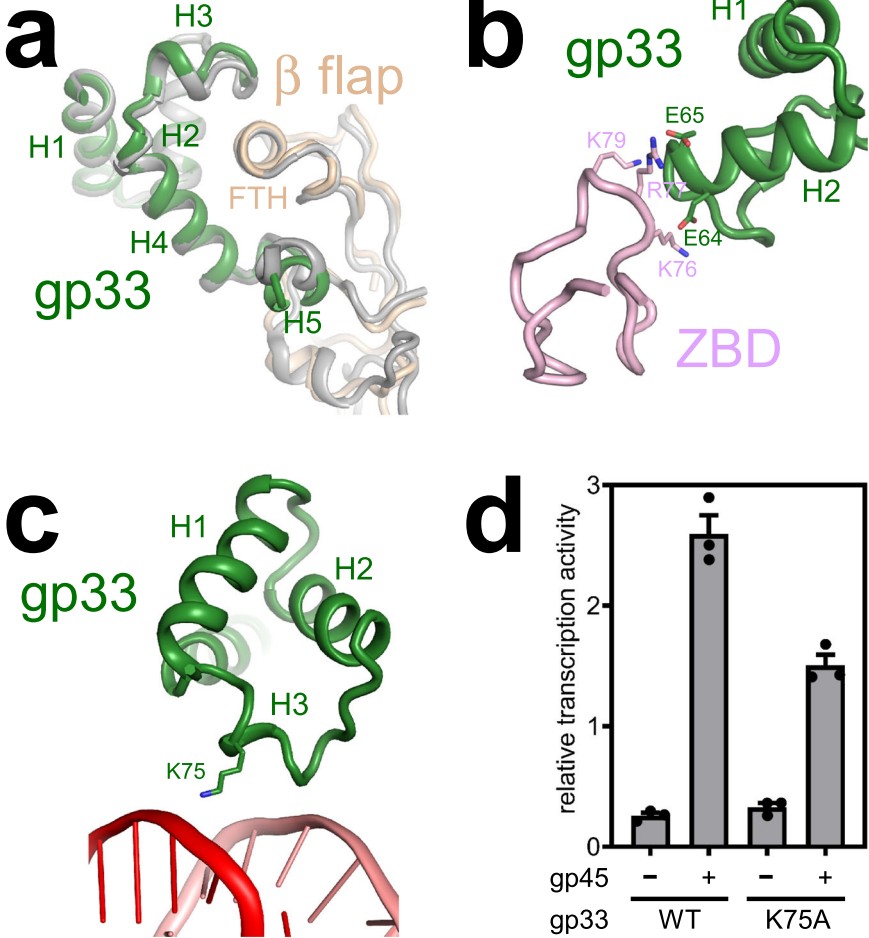

**Fig. 5 Interactions of gp33 with RNAP and the upstream dsDNA. a** Gp33 clamps onto the β flap tip helix (FTH). Dark green, gp33 in TAC-gp45; wheat, the β flap in TAC-gp45; gray, the crystal structure of gp33 complexed with the β flap (PDB 3TBI [https://doi.org/10.2210/pdb3TBI/pdb]). **b** A patch of negatively charged residues on gp33 is positioned near a patch of positively charged residues on the zinc-binding domain (ZBD). Dark green, gp33; light pink, ZBD. **c** Gp33 residue K75 is positioned to make an electrostatic interaction with the DNA backbone phosphate. Dark green, gp33; salmon, nontemplate-strand DNA; red, template-strand DNA. **d** Alanine substitution of gp33 residue K75 affects gp45-dependent transcription activation, but not gp33-dependent transcription repression. Error bars represent mean ± SEM out of $n = 3$ experiments.

substitution of E64 and E65 in combination decreases gp33–RNAP interaction in competitive binding experiments[15].

**Interactions between gp33 and the upstream dsDNA.** In our structure, the H3 of gp33 contacts the upstream dsDNA. In particular, residue K75 is positioned to form a salt bridge with the DNA backbone phosphate (Fig. 5c and Supplementary Fig. 5f). Gp33 mutant protein (K75A) represses basal transcription as efficiently as wild-type protein (Fig. 5d), indicating that the mutant protein is well folded and the gp33–DNA interaction is not required for transcription repression. However, the mutant protein activates transcription to a less extent than wild-type protein, indicating that the gp33–DNA interaction is important for transcription activation. In addition, gp33 contacts the DNA backbone only, consistent with the observation that the sequences of the upstream dsDNA are not conserved in T4 late promoters[23].

**Interactions between gp45 and the SCBMs.** A triangular density feature adjacent to gp33 can be attributed to gp45 homotrimer (Supplementary Fig. 5g). For convenience, the three protomers (gp45[I], gp45[II], and gp45[III]) are numbered clockwise, looking from downstream to upstream (Fig. 6a). The upstream dsDNA goes through the center of the closed ring of gp45. The lateral face with the protruding C-terminus presents a hydrophobic patch on each

protomer that serves as a binding site for the SCBMs of DNA polymerase, DNA ligase, gp33, and gp55[18,24]. Unfortunately, the local resolution of gp45 is too low to build the model for the SCBMs of gp33 and gp55 (Supplementary Fig. 4c). To determine which protomers gp33 and gp55 most likely bind to, we super-impose the crystal structure of gp45 in complex with the SCBM of T4 DNA ligase[24] on the cryo-EM structure of TAC-gp45. The distance between the C-terminus of the gp33 structure (residue 102) and the N-terminus of the SCBM (corresponding to residue 106 of gp33 according to Supplementary Table 2) bound to gp45[I] is 13 Å, which is within the range of distances that could be spanned by a three-residue linker. Therefore, the SCBM of gp33 probably binds to gp45[I]. The distances between the C-terminus of the gp55 structure (residue 153) and the N-termini of the SCBMs (corresponding to residue 175 of gp55 according to Supplementary Table 2) bound to gp45[II] and gp45[III] are 45 and 71 Å, respectively, which are within the range of distances that could be spanned by a 21-residue linker, but the linker needs to bypass the upstream dsDNA in the case of gp45[III] (Fig. 6b). Therefore, the SCBM of gp55 probably binds to gp45[II].

## Discussion
In this work, we determine the precise structures of T4 late gene transcription machinery. The structures unequivocally show that

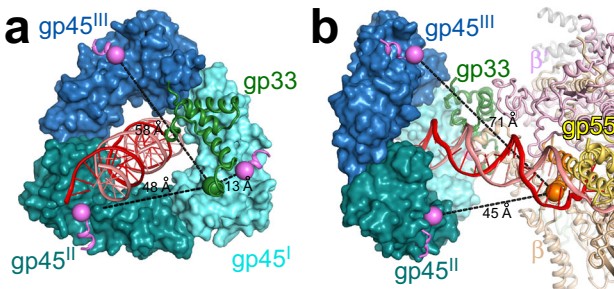

**Fig. 6 Interactions of gp45 with the SCBMs. a** The distances between gp33 and the putative SCBMs. The crystal structure of gp45 complexed with the SCBM of T4 DNA ligase (PDB 6DRT [https://doi.org/10.2210/pdb6DRT/pdb]) is superimposed on TAC-gp45. View orientation, from downstream to upstream. The C-terminus of the gp33 structure and the N-termini of the SCBMs are shown as spheres. Magenta, SCBM of T4 DNA ligase. Other colors as in Fig. 1c. **b** The distances between gp55 and the putative SCBMs. The C-terminus of the gp55 structure and the N-termini of the SCBMs are shown as spheres. Magenta, SCBM of T4 DNA ligase. Other colors as in Fig. 1c.

a sliding clamp can turn on transcription by physically tethering RNAP to the DNA. Our structures confirm that gp55 is a diverged σ[70] family protein, contacting the clamp helices and mediating the sequence-specific interactions with the promoter -10-like element. In addition to the σR2 homology domain, we find that gp55 has a HLH motif, which is critical for T4 late transcription. The structure of TAC-gp45 reveals that gp33 contacts the upstream dsDNA, which is important for gp45-depenent transcription activation. The structure verifies that gp45 is located at the upstream end of TAC-gp45, in the vicinity of gp33. The gp45 protomers, to which gp33 and gp55 bind, are assigned based on the restraints of their linkers.

A model of gp45-dependent transcription activation has been proposed based on the elegant biochemical studies by Geiduschek and Kassavetis (Fig. 7)[23]. To initiate transcription, RNAP must locate promoter sequences, which compose a very small amount of the genome. Single-molecule experiments have demonstrated that the promoter search is dominated by three-dimensional diffusion[25,26]. As for bacteriophage T4 late gene transcription, ATP hydrolysis by gp44/gp62 complex triggers the detachment of gp45 from the gp44/gp62 complex, freeing gp45 to track along DNA. RNAP–gp55–gp33 grabs gp45 through the SCBMs of gp33 and gp55, and travels along with gp45. This kind of one-dimensional search is intuitively more efficient than three-dimensional diffusion.

The cryo-EM structure of TAC-gp45 shows that the HLH motif of gp55 blocks the exit path for the nascent RNA (Supplementary Fig. 7) and must be displaced to allow further extension of the RNA and transition into transcription elongation complex. After gp55 is displaced, the general transcription factor NusG could bind to the clamp helices and enhance elongation by suppressing RNAP backtracking[27]. The cryo-EM structure further shows that gp33 and gp45 would not occlude the RNA exit channel and could remain bound to RNAP during elongation. Even though gp55 must be displaced from RNAP, it could remain tethered to gp45 through the SCBM. Therefore, the quaternary complex resumes promoter scanning immediately after transcription termination, which may further boost T4 late gene transcription. The binding of gp33 during elongation would decrease the accessibility of the FTH to the general transcription factor NusA, which plays key roles in ρ-dependent transcription termination and transcription-translation coupling[28–30]. Therefore, both processes would be compromised during T4 late

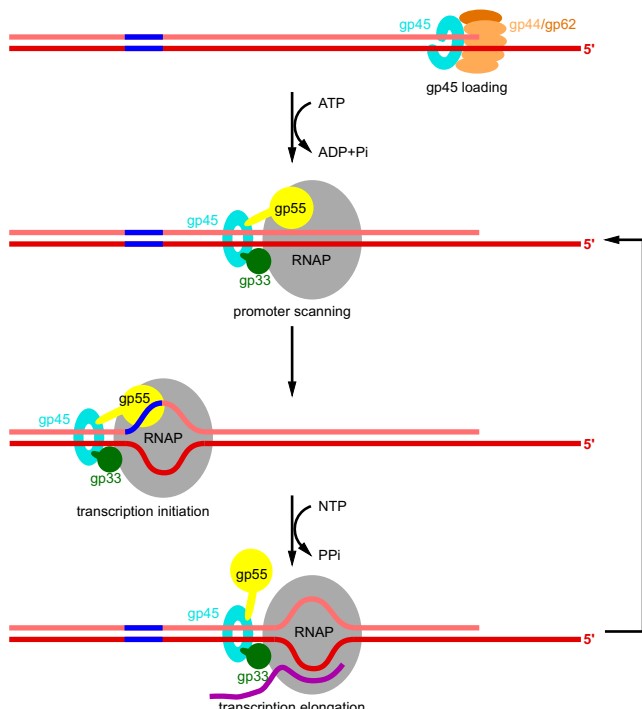

**Fig. 7 Proposed model of gp45-dependent transcription activation.** Gp45 is mounted onto DNA by gp44/gp62 complex at double-strand-single-strand junctions. Then, RNAP is tethered to gp45 through gp55 and gp33. The quaternary complex scans for promoters along the DNA. If a T4 late promoter is encountered, the quaternary complex unwinds the promoter DNA and starts RNA synthesis. During transcription elongation, gp55 is displaced by the nascent RNA, but remains tethered to gp45. After transcription termination, the quaternary complex resumes promoter scanning without dissociation from the DNA. Gray, RNAP; light orange, gp44; dark orange, gp62; cyan, gp45; yellow, gp55; dark green, gp33; salmon, nontemplate-strand DNA; red, template-strand DNA; blue, -10-like element; magenta, RNA.

transcription. Although our structures are consistent with the model that the quaternary complex stays together during elongation, we cannot rule out the possibility that gp33, gp55, and gp45 might fall off RNAP during elongation.

Transcription activation is a ubiquitous strategy of transcription regulation and it generally involves activators. Most activators are DNA-binding proteins that bind to enhancers or promoter-proximal elements[31,32]. Instead of using DNA-binding activators, T4 bacteriophage activates the transcription of late genes by using a DNA-tracking protein, the sliding clamp, as the activator. As a proof of concept, fusion proteins linking the transcription activation domain of herpes simplex virus VP16 protein to a sliding clamp have been shown to activate transcription by yeast and *Drosophila* RNAP II in vitro[33]. It is likely that other instances of the use of this simple mechanism for transcription activation would be found in nature.

## Methods

**Gp55**. Gp55 was prepared as described[13]. Briefly, *E. coli* strain BL21(DE3) (Invitrogen, Inc.) was transformed with plasmid pET21a-gp55 (GENEWIZ, Inc.) encoding gp55 under the control of the bacteriophage T7 gene *10* promoter. Single colonies of the resulting transformants were used to inoculate 1 l LB broth containing 100 μg/ml ampicillin, cultures were incubated at 37 °C with shaking until $OD_{600} = 0.9$, cultures were induced by addition of IPTG to 1 mM, and cultures were incubated 17 h at 16 °C. Then, cells were harvested by centrifugation (5422 × *g*; 15 min at 4 °C), resuspended in 20 ml buffer A (50 mM Tris-HCl, pH 7.9, 0.1 M NaCl, and 5% glycerol), and lysed using a JN-02C cell disrupter (JNBIO, Inc.). After centrifugation (24,792 × *g*; 40 min at 4 °C), the pellet was washed with 25 ml

buffer A supplemented with 0.25% Triton X-100 twice, resuspended in buffer B (50 mM Tris-HCl, pH 7.9, 6 M GuHCl, 1 mM EDTA, and 10% glycerol), and dialyzed against 1 l buffer C (20 mM Tris-HCl, pH 7.9, 0.2 M NaCl, 1 mM EDTA, and 5 mM β-mercaptoethanol) for three times. After centrifugation (24,792 × $g$; 30 min at 4 °C), the supernatant was diluted by buffer D (20 mM Tris-HCl, pH 7.9, 1 mM EDTA, 1 mM DTT, 5% glycerol), loaded onto a Mono Q 10/100 GL column (GE Healthcare, Inc.) equilibrated in buffer D, and eluted with a 160 ml linear gradient of 0–1 M NaCl in buffer D. Fractions containing gp55 were concentrated to 5 ml using an Amicon Ultra-15 Centrifugal Filter (10 kDa MWCO; Merck Millipore, Inc.) and applied to a HiLoad 16/600 Superdex 200 column (GE Healthcare, Inc.) equilibrated in buffer E (10 mM HEPES, pH 7.5, 0.05 M KCl), and the column was eluted with 120 ml of the same buffer. Gp55 derivatives were expressed and purified in the same way as wild-type protein.

**Gp33**. Gp33 was prepared as described[13]. Briefly, *E. coli* strain BL21(DE3) (Invitrogen, Inc.) was transformed with plasmid pET28a-NH-gp33 (GENEWIZ, Inc.) encoding N-hexahistidine-tagged gp33 under the control of the bacteriophage T7 gene *10* promoter. Single colonies of the resulting transformants were used to inoculate 5 l LB broth containing 50 μg/ml kanamycin, cultures were incubated at 37 °C with shaking until $OD_{600} = 0.7$, cultures were induced by addition of IPTG to 1 mM, and cultures were incubated 3 h at 37 °C. Then, cells were harvested by centrifugation (5422 × $g$; 10 min at 4 °C), resuspended in 75 ml lysis buffer (20 mM Tris-HCl, pH 8.0, 0.5 M NaCl, 5% glycerol), and lysed using a JN-02C cell disrupter (JNBIO, Inc.). The lysate was centrifuged (24,792 × $g$; 40 min at 4 °C), and the supernatant was loaded onto a 2 ml column of Ni-NTA agarose (Qiagen, Inc.) equilibrated with lysis buffer. The column was washed with 10 ml lysis buffer containing 0.04 M imidazole and eluted with 10 ml lysis buffer containing 0.16 M imidazole. The sample was further purified by anion-exchange chromatography on a Mono Q 10/100 GL column (GE Healthcare, Inc.; 160 ml linear gradient of 0.1–1 M NaCl in buffer D). Fractions containing gp33 were concentrated to 5 ml using an Amicon Ultra-15 centrifugal filter (10 kDa MWCO; Merck Millipore, Inc.) and applied to a HiLoad 16/600 Superdex 200 column (GE Healthcare, Inc.) equilibrated in buffer E, and the column was eluted with 120 ml of the same buffer. Gp33 derivatives were expressed and purified in the same way as wild-type protein.

**Gp45**. Gp45 was prepared as described[24]. Briefly, *E. coli* strain BL21(DE3) (Invitrogen, Inc.) was transformed with plasmid pET24a-gp45-CH (GENEWIZ, Inc.) encoding C-hexahistidine-tagged gp45 under the control of the bacteriophage T7 gene *10* promoter. Single colonies of the resulting transformants were used to inoculate 1 l LB broth containing 50 μg/ml kanamycin, cultures were incubated at 37 °C with shaking until $OD_{600} = 0.9$, cultures were induced by addition of IPTG to 1 mM, and cultures were incubated 17 h at 16 °C. Then, cells were harvested by centrifugation (5422 × $g$; 10 min at 4 °C), resuspended in 25 ml buffer A, and lysed using a JN-02C cell disrupter (JNBIO, Inc.). The lysate was centrifuged (24,792 × $g$; 40 min at 4 °C), and the supernatant was loaded onto a 2 ml column of Ni-NTA agarose (Qiagen, Inc.) equilibrated with buffer A. The column was washed with 10 ml buffer A containing 0.08 M imidazole and eluted with 10 ml buffer A containing 0.5 M imidazole. The eluate was concentrated to 5 ml using an Amicon Ultra-15 centrifugal filter (10 kDa MWCO; Merck Millipore, Inc.) and applied to a HiLoad 16/600 Superdex 200 column (GE Healthcare, Inc.) equilibrated in buffer E, and the column was eluted with 120 ml of the same buffer.

**Gp44/gp62 complex**. Gp44/gp62 complex was prepared as described[34]. Briefly, *E. coli* strain BL21(DE3) (Invitrogen, Inc.) was transformed with plasmid pET24a-gp44-gp62-CH (GENEWIZ, Inc.) encoding gp44 and C-hexahistidine-tagged gp62 under the control of the bacteriophage T7 gene *10* promoter. Single colonies of the resulting transformants were used to inoculate 2 l LB broth containing 50 μg/ml kanamycin, cultures were incubated at 37 °C with shaking until $OD_{600} = 1.1$, cultures were induced by addition of IPTG to 1 mM, and cultures were incubated 15 h at 16 °C. Then, cells were harvested by centrifugation (5422 × $g$; 10 min at 4 °C), resuspended in 55 ml buffer F (20 mM Tris-HCl, pH 7.5, 0.3 M NaCl, and 5% glycerol), and lysed using a JN-02C cell disrupter (JNBIO, Inc.). The lysate was centrifuged (24,792 × $g$; 40 min at 4 °C), and the supernatant was loaded onto a 2 ml column of Ni-NTA agarose (Qiagen, Inc.) equilibrated with buffer F. The column was washed with 10 ml buffer F containing 0.04 M imidazole and eluted with 10 ml buffer F containing 0.5 M imidazole. The eluate was concentrated to 5 ml using an Amicon Ultra-15 centrifugal filter (10 kDa MWCO; Merck Millipore, Inc.) and applied to a HiLoad 16/600 Superdex 200 column (GE Healthcare, Inc.) equilibrated in buffer E, and the column was eluted with 120 ml of the same buffer.

**Gp32**. *E. coli* strain BL21(DE3)pLysS (Invitrogen, Inc.) was transformed with plasmid pET21a-gp32 (GENEWIZ, Inc.) encoding gp32 under the control of the bacteriophage T7 gene *10* promoter. Single colonies of the resulting transformants were used to inoculate 5 l LB broth containing 50 μg/ml ampicillin, cultures were incubated at 37 °C with shaking until $OD_{600} = 0.8$, cultures were induced by addition of IPTG to 0.4 mM, and cultures were incubated 3 h at 37 °C. Then, cells were harvested by centrifugation (5422 × $g$; 10 min at 4 °C), resuspended in 70 ml buffer G (10 mM Tris-HCl, pH 7.5, 0.1 M NaCl, 5% glycerol, 1 mM EDTA, and 1 mM DTT), and lysed using a JN-02C cell disrupter (JNBIO, Inc.). The lysate was

centrifuged (24,792 × $g$; 45 min at 4 °C), and the supernatant was loaded onto a 5 ml column of HiTrap Heparin HP (GE Healthcare, Inc.) equilibrated in buffer G and eluted with a 100 ml linear gradient of 0.1–1 M NaCl in buffer G. The sample was further purified by anion-exchange chromatography on a Mono Q 10/100 GL column (GE Healthcare, Inc.; 160 ml linear gradient of 0.1–1 M NaCl in buffer G). Fractions containing gp32 were concentrated to 5 ml using an Amicon Ultra-15 centrifugal filter (10 kDa MWCO; Merck Millipore, Inc.) and applied to a HiLoad 16/600 Superdex 200 column (GE Healthcare, Inc.) equilibrated in buffer E, and the column was eluted with 120 ml of the same buffer.

**E. coli RNAP**. *E. coli* RNAP was prepared from *E. coli* strain BL21(DE3) (Invitrogen, Inc.) transformed with plasmid pIA900[35]. Single colonies of the resulting transformants were used to inoculate 50 ml LB broth containing 100 μg/ml ampicillin, and cultures were incubated 16 h at 37 °C with shaking. Aliquots (10 ml) were used to inoculate 1 l LB broth containing 100 μg/ml ampicillin, cultures were incubated at 37 °C with shaking until $OD_{600} = 0.6$, cultures were induced by addition of IPTG to 1 mM, and cultures were incubated 3 h at 37 °C. Then, cells were harvested by centrifugation (5422 × $g$; 15 min at 4 °C), resuspended in 20 ml lysis buffer (50 mM Tris-HCl, pH 7.9, 0.2 M NaCl, 2 mM EDTA, 5% glycerol, and 5 mM DTT), and lysed using a JN-02C cell disrupter (JNBIO, Inc.). After poly(ethyleneimine) precipitation and ammonium sulfate precipitation, the pellet was resuspended in buffer H (10 mM Tris-HCl, pH 7.9, 0.5 M NaCl, and 5% glycerol) and loaded onto a 5 ml column of Ni-NTA agarose (Qiagen, Inc.) equilibrated with buffer H. The column was washed with 25 ml buffer H containing 20 mM imidazole and eluted with 25 ml buffer H containing 0.15 M imidazole. The eluate was diluted in buffer F and loaded onto a Mono Q 10/100 GL column (GE Healthcare, Inc.) equilibrated in buffer F and eluted with a 160 ml linear gradient of 0.3–0.5 M NaCl in buffer F. Fractions containing *E. coli* RNAP core enzyme were pooled and stored at −80 °C.

**Nucleic acid scaffold**. Double-stranded, 3′-end recessed DNA scaffold were prepared by Exo III treatment as described[13]. Briefly, an ~600-bp PCR fragment containing the $P_{23}$ promoter, the mango III coding sequence, and *rrnB* terminators amplified from plasmid pUC57-$P_{23}$-mango (GENEWIZ, Inc.) using primers in Supplementary Table 3 was cleaved with Hind III and Kpn I to generate Exo III-susceptible downstream and Exo III-resistant upstream ends and reacted with Exo III to generate ~150 nt of ssDNA.

**Mango III transcription assay**. Mango III transcription assay was performed in a 384-well microplate format. Reaction mixtures contained (50 μl): 0.2 μM *E. coli* RNAP, 0.02 μM nucleic acid scaffold, 0 or 0.4 μM gp55, 0 or 0.4 μM gp33, 0 or 0.4 μM gp45 homotrimer, 0.1 μM gp44/gp62 complex, 3.75 μM gp32, 2 mM dATP, 0.1 mg/ml heparin, 1 mM ATP, 1 mM GTP, 1 mM UTP, 1 mM CTP, and 1 μM TO1-biotin in transcription buffer (33 mM Tris-acetate, pH 7.9, 0.2 M potassium acetate, 10 mM magnesium acetate, and 1 mM DTT). *E. coli* RNAP was incubated with gp55, gp33, and gp45 for 10 min at 22 °C, incubated with DNA scaffold, gp44/gp62 complex, gp32, and dATP for 10 min at 22 °C, incubated with heparin for 1 min at 22 °C, and incubated with NTPs and TO1-biotin for 20 min at 37 °C. Fluorescence emission intensities were measured using a Varioskan Flash Multimode Reader (Thermo Fisher, Inc., excitation wavelength = 510 nm; emission wavelength = 535 nm).

**Assembly of RPo-gp55 and TAC-gp45**. RPo-gp55 and TAC-gp45 were prepared in reaction mixtures containing (30 μl): 0.8 μM *E. coli* RNAP, 0.2 μM nucleic acid scaffold, 0.8 μM gp55, 0.8 μM gp33, 0.8 μM gp45 homotrimer, 0.2 μM gp44/gp62 complex, 3.75 μM gp32, and 2 mM dATP in transcription buffer. *E. coli* RNAP was incubated with gp55, gp33, and gp45 for 10 min on ice, and incubated with DNA scaffold, gp44/gp62 complex, gp32, and dATP for 20 min at 30 °C.

**Cryo-EM grid preparation**. Quantifoil grids (R 1.2/1.3 Cu 300 mesh; Quantifoil, Inc.) were glow-discharged for 120 s at 25 mA prior to the application of 3 μl of the complex in transcription buffer, then plunge-frozen in liquid ethane using a Vitrobot (FEI, Inc.) with 95% chamber humidity at 10 °C.

**Cryo-EM data acquisition and processing**. The grids were imaged using a 300 kV Titan Krios (FEI, Inc.) equipped with a K2 Summit direct electron detector (Gatan, Inc.). Images were recorded with Serial EM[36] in counting mode with a physical pixel size of 1.307 Å and a defocus range of 1.5-2.5 μm. Data were collected with a dose of 10 e/pixel/s. Images were recorded with a 10 s exposure and 0.25 s subframes to give a total dose of 59 e/Å². Subframes were aligned and summed using MotionCor2[37]. The contrast transfer function was estimated for each summed image using CTFFIND4[38]. From the summed images, ~10,000 particles were manually picked and subjected to 2D classification in RELION[39]. 2D averages of the best classes were used as templates for auto-picking in RELION. Auto-picked particles were manually inspected, then subjected to 2D classification in RELION. Poorly populated classes were removed. These particles were 3D classified in RELION using a map of *E. coli* TEC (EMD-8585 [https://www.emdataresource.org/EMD-8585])[40] low-pass filtered to 40 Å resolution as a reference. Initial 3D

classification resulted in a reconstruction with densities for gp45 (TAC-gp45) and a reconstruction without densities for gp45 (RPo-gp55). For RPo-gp55, further 3D classification with four classes was performed. The density map of the upstream dsDNA of class 1 is weak compared with the other three classes. Therefore, only classes 2–4 were combined for refinement. For TAC-gp45, masked 3D classification leads to four classes with slightly different conformations of gp45 relative to RNAP, which is consistent with the model that gp45 is tethered to RNAP through flexible linkers. Class 3 is refined because its resolution is the highest among the four classes. CTF refinement and particle polishing were performed before final 3D refinement and postprocessing.

**Cryo-EM model building and refinement.** The homology model of gp55 was generated on Phyre2 server[41]. The model of RNAP from the structure of RPo-σ[70] (PDB 6CA0 [https://doi.org/10.2210/pdb6CA0/pdb])[6], the homology model of gp55, and the crystal structures of gp33 and gp45 were fitted into the cryo-EM density map using Chimera[42]. The model of nucleic acids was built manually in Coot[43]. The coordinates were real-space refined with secondary structure restraints in Phenix[44].

**Pull-down assay.** Pull-down assays were performed, taking advantage of the fact that the β′ subunit of *E. coli* RNAP is C-terminally His-tagged. Overall, 250 μl *E. coli* RNAP (1 μM) in buffer F was incubated with 50 μl Ni-NTA agarose (Qiagen, Inc.) for 5 min at 22 °C. After the resin was centrifuged (9391 × g; 2 min at 22 °C) and washed with 0.5 ml buffer F for three times, 250 μl gp55 or gp55 derivatives (1 μM) in buffer F was added to the resin and incubated for 5 min at 22 °C. After the resin was centrifuged (9391 × g; 2 min at 22 °C) and washed with 0.5 ml buffer F for three times, bound proteins were analyzed by SDS-PAGE and quantified by ImageJ (https://imagej.nih.gov/ij/).

**Electrophoretic mobility shift assay.** Template-strand DNA (5′-AGTCCGGCT GCTTCAGTATCAGGAGTATTTATACTCTCAGTAATAGTGCTGAGCTCTTT ATTAG-3′, Sangon Biotech, Inc.) and nontemplate-strand DNA (5′-CTAATAAA GAGCTCAGCACTATTACTGAGAGTATAAATACTCCTGATACTGAAG CAGCCGGACT-3′, Sangon Biotech, Inc.) were annealed at a 1:1 ratio in 10 mM Tris-HCl, pH 7.9, 0.2 M NaCl, and stored at −80 °C. Electrophoretic mobility shift assays were performed in reaction mixtures containing (10 μl): 0.4 μM gp55 or gp55 derivatives, 0.2 μM *E. coli* RNAP, 0.2 μM DNA scaffold, 0.1 mg/ml heparin, and 5% glycerol in buffer F. *E. coli* RNAP was incubated with gp55 or gp55 derivatives for 10 min on ice, incubated with DNA scaffold for 20 min at 37 °C, and incubated with 0.1 mg/ml heparin for 1 min at 22 °C. The reaction mixtures were applied to 5% polyacrylamide slab gels (29:1 acrylamide/bisacrylamide), electrophoresed in 90 mM Tris-borate, pH 8.0, and 0.2 mM EDTA, stained with 4S Red Plus Nucleic Acid Stain (Sangon Biotech, Inc.) according to the procedure of the manufacturer.

**Reporting summary.** Further information on research design is available in the Nature Research Reporting Summary linked to this article.

## Data availability

The accession codes for the cryo-EM density maps reported in this paper are Electron Microscopy Data Bank: EMD-30604 (RPo-gp55) and EMD-30605 (TAC-gp45). The accession codes for the atomic coordinates reported in this paper are Protein Data Bank: 7D7C (RPo-gp55) and 7D7D (TAC-gp45). The accession code for the cryo-EM density map used in this study is EMD-8585. The accession codes for the atomic coordinates used in this study are PDB 6CA0, PDB 3TBI, and PDB 6DRT. Source data are provided with this paper. Other data are available from the corresponding author upon reasonable request.

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

## Acknowledgements

The authors would like to thank Shenghai Chang at the Center of Cryo Electron Microscopy in Zhejiang University School of Medicine for help with cryo-EM data collection. The authors would like to thank the Core Facilities, Zhejiang University School of Medicine, for the technical support. This work was funded by the National Natural Science Foundation of China (31970040 to Y.F. and 32000025 to J.S.).

## Author contributions

J.S., A.W., S.J., B.G., and Y.H. performed the experiments. Y.F. supervised the experiments and wrote the manuscript with contributions from the other authors. All authors contributed to the analysis of the data and the interpretation of the results.

## Competing interests

The authors declare no competing interests.
