## [Peer Review File · Nature Communications]

REVIEWER COMMENTS

Reviewer #1 (Remarks to the Author):

Recommendation: Reject, manuscript writing, and figure drawing are poor. The experimental method provides insufficient details, and the EM data processing is incomplete.

Reviewer's comments:

Shi, Wen, and Jin et al. determined cryo-EM structures of T4 gp55 bound EcoRNAP open complex and gp45-bound EcoRNAP transcription activation complex at 3.6 and 4.5 Å, respectively. T4 phage gp55 and gp33 are the first discovered master regulators of the viral developmental program, and gp55 is the smallest and divergent member of $\sigma 70$. From these structures, the authors reveal the interactions between gp55 and the promoter DNA, explaining how T4 late promoters, which contain non-canonical -10 elements, are recognized. Besides the $\sigma R2$ -homologous region of gp55, the gp55 HLH motif aids RNAP binding. Also, the authors revealed GP45 encircles the DNA and tethers RNAP to it, supporting the idea that gp45 switches the promoter search from 3D diffusion mode to 1D scanning mode.

First of all, the manuscript is poorly written, and the figures are not well-drawn, so it was not easy to catch what the text is telling. For example, the authors named the two cryo-EM structures as 'Rpo-gp55' and 'TAC-gp45' in line 76 but did not explain why one is RPo, and another is TAC. In line 95, the authors stated, 'we incubate all components necessary for gp-45-dependent transcription activation', which is not appropriate for scientific writing. All the principal components should be listed here, as far as this reviewer learned. The manuscript uses many ambiguous sentences, which make the manuscript incomprehensible. Also, the referenced information is fragmentary - without logical connection with prior or posterior sentences. For figures, for example, the subunits of RNA polymerase are not indicated in figure 1b. The main components – T4 gp55, gp45, gp33 – are not clearly shown in the figure as well.

In the method section, the authors even did not include the buffer conditions they used for cryo-EM grid preparation, and many details of the experimental sections are omitted. Also, in this reviewer's opinion, current cryo-EM data processing is likely insufficient. Particle polishing, which is almost mandatory now, was not performed.

This research made a few discoveries, but these were not well described in the context of transcription research and insufficient to make the manuscript publishable in Nature Communication.

Reviewer #2 (Remarks to the Author):

This paper reports the EM-obtained structures of two aspects of the T4 late gene transcription machinery: 1) the open complex of RNA polymerase with the T4 late sigma factor, gp55, and the promoter template and 2) the activated complex of RNAP with gp55, the T4 coactivator (the sigma70 Region 4 substitute gp33), the sliding clamp enhancer (gp45), and the DNA. It is truly exciting to now be able to visualize these complexes, and this is a very nice contribution to the field. The visualized complexes essentially confirm every aspect of the machinery that was worked out through the

elegant biochemistry by George Kassavetis and others in the lab of Peter Geiduschek.

My main concern is the tone of the paper. Although the authors do reference the Geiduschek work, they do not really reflect how their work confirms what is already known. For example, the model in Fig.6 is exactly the same as what already appears in reviews discussing this type of transcription, all built from the previous biochemistry. Furthermore, the authors state in line 69, 'Genetic and biochemical studies have elucidated some aspects of gp45-dependent transcription activation. Nevertheless, a precise mechanistic understanding of the process remains elusive, in part, because of a lack of structural information for an intact gp45-dependent transcription activation complex.' Perhaps this is true in a very technical sense, but it certainly undervalues the work that has been done previously. In addition, because of their lack of density in the SCBM, even they are not able to add these details at this time. A novice or young reader would assume from the way the paper is written that their structural analyses are providing the basis for the model in Fig. 6.

I want to be clear that I am not suggesting that this paper is not an excellent contribution. I am instead saying that the work would be even more powerful, if the authors had chosen to change their point-of-view and acknowledge specifically and in detail what has been done, what was known, and what they either confirmed or found as new. It does not diminish this work to indicate how it follows from others.

Other points:

- 1) I am confused why the authors chose to use the Mango readout to test the transcription activities of their proteins. Why didn't they observe transcription directly? The level of enhancement observed in Fig. S1C in the presence of complete system with the enhancer is low. Do the authors have an explanation for that? They should point out how their level of enhancement compares with previous work.
- 2) Fig. S2 needs much more information. What were the other 75% of the particles with the TAC-gp45 molecule that were not refined? What was class I, which was not refined for the RPo? How did they determine which class to use?
- 3) An input control is needed for Fig. 2E – Accounting for the smaller size of the gp55 truncated protein at 114 aa, it is not clear to me that the binding is significantly affected. Please quantify and show std deviation.

Reviewer #3 (Remarks to the Author):

Activation of late phage T4 transcription is an important and interesting phenomenon, as it occurs by a distinct mechanism, different from other transcription activatory mechanisms. This is a clearly presented piece of structural biology that, to my knowledge, breaks new ground, and it finishes with a convincing model.

Two issues trouble me:

1. does this mode of activation apply to other systems?
2. does the RNAP/gp33/gp55/gp45 quaternary complex really stay together during elongation?

To my mind, more discussion is needed concerning both points

Reviewer #1 (Remarks to the Author):

Shi, Wen, and Jin et al. determined cryo-EM structures of T4 gp55 bound EcoRNAP open complex and gp45-bound EcoRNAP transcription activation complex at 3.6 and 4.5 Å, respectively. T4 phage gp55 and gp33 are the first discovered master regulators of the viral developmental program, and gp55 is the smallest and divergent member of $\sigma 70$. From these structures, the authors reveal the interactions between gp55 and the promoter DNA, explaining how T4 late promoters, which contain non-canonical -10 elements, are recognized. Besides the $\sigma R2$ -homologous region of gp55, the gp55 HLH motif aids RNAP binding. Also, the authors revealed GP45 encircles the DNA and tethers RNAP to it, supporting the idea that gp45 switches the promoter search from 3D diffusion mode to 1D scanning mode.

First of all, the manuscript is poorly written, and the figures are not well-drawn, so it was not easy to catch what the text is telling. For example, the authors named the two cryo-EM structures as 'RPO-gp55' and 'TAC-gp45' in line 76 but did not explain why one is RPO, and another is TAC. In line 95, the authors stated, 'we incubate all components necessary for gp-45-dependent transcription activation', which is not appropriate for scientific writing. All the principal components should be listed here, as far as this reviewer learned. The manuscript uses many ambiguous sentences, which make the manuscript incomprehensible. Also, the referenced information is fragmentary - without logical connection with prior or posterior sentences. For figures, for example, the subunits of RNA polymerase are not indicated in figure 1b. The main components – T4 gp55, gp45, gp33 – are not clearly shown in the figure as well.

We thank the reviewer for the constructive suggestions to improve the quality of the manuscript. The text and figures have been improved in the revised version. For example, all the principal components have been listed in line 111. The subunits of RNAP have been labelled in Figure 1. To show the main components clearly, the density maps have been omitted from Figure 1 and presented in Figure S5.

TAC is activator bound RPO. TAC-gp45 is named TAC to emphasize that it is different from RPO-gp55, which does not comprise a transcription activator. The definitions of RPO-gp55 and TAC-gp45 have been added in lines 60 and 69, respectively.

In the method section, the authors even did not include the buffer conditions they used for cryo-EM grid preparation, and many details of the experimental sections are omitted. Also, in this reviewer's opinion, current cryo-EM data processing is likely insufficient. Particle polishing, which is almost mandatory now, was not performed.

The details of the experimental sections have been added in the revised version. Particle polishing, which was actually performed, has been emphasized in line 424.

This research made a few discoveries, but these were not well described in the context of

transcription research and insufficient to make the manuscript publishable in Nature Communication.

We have rewritten the Introduction to describe the previous work, and our findings have been put into context in the Results and Discussion. The added information is excerpted below:

Bacteriophages use bacterial RNAP to transcribe their own genes. For decades, transcription of bacteriophage T4 late genes has served as a model system to investigate mechanisms of transcription regulation. The T4 late promoters consist of a -10-like element placed at the position corresponding to the bacterial promoter -10 element ⁸. Nevertheless, there is no sequence conservation at the position corresponding to the bacterial promoter -35 element. Peter Geiduschek and colleagues developed an *in vitro* system that leads to the current understanding of T4 late transcription ⁹. It was found that T4 encoded gp55 enabled RNAP to execute low level (basal) transcription, which was further activated by the sliding clamp gp45 and its coactivator gp33.

Gp55, a highly diverged σ^{70} family protein ¹⁰, plays an essential role in T4 late transcription. To initiate basal transcription, bacterial RNAP needs to form holoenzyme with gp55, which recognizes T4 late promoters and confers the ability to form a catalytically competent RNAP-promoter open complex (RPO-gp55) ¹¹. The RNAP binding motif of gp55 has been inferred on the basis of alanine scanning mutagenesis analyzed for RNAP binding, basal and activated transcription ¹². Initial binding of RNAP-gp55 holoenzyme to DNA is not sequence-specific, while RPO-gp55 is sequence-specific and readily detected by footprinting ^{13,14}. The acquisition of sequence specificity on promoter opening implies recognition of some feature of the transcription bubble by gp55, but this has not been demonstrated directly.

Gp33 and gp45 further activate the basal transcription activity of RNAP-gp55 holoenzyme by forming a catalytically competent transcription activation complex (TAC-gp45) ¹⁴. Pull-down experiments showed that gp33 binds to the RNAP β flap ¹⁵, which was further confirmed by the crystal structure of gp33 complexed with the RNAP β flap ¹⁶. Although gp33 does not, by itself, bind to DNA ⁹, photo-crosslinking experiments showed that gp33 was proximal to the upstream double-stranded DNA (dsDNA) in TAC-gp45 ^{14,17}. Without a precise structure of TAC-gp45, whether gp33 directly contacts the upstream dsDNA is unknown. DNase I footprinting and photo-crosslinking experiments collectively suggested that gp45 was located at the upstream end of TAC-gp45, in the vicinity of the coactivator, gp33 ¹⁴. Gp45 is a homotrimer, forming a triangle with a central hole large enough to accommodate dsDNA ¹⁸. The lateral face with the protruding C-terminus presents a hydrophobic patch on each protomer that serves as a binding site for the sliding clamp binding motif (SCBM) that is attached to the body of gp55 and gp33 through a linker. *In vitro* transcription assays with SCBM truncated proteins showed that the SCBM of gp33 is essential for transcription activation, while loss of the SCBM of gp55 impairs but does not abolish transcription activation ^{19,20}.

For the first time, we determine the precise structures of T4 late gene transcription machinery. The structures unequivocally show that a sliding clamp can turn on transcription by physically tethering RNAP to the DNA. Our structures confirm that gp55 is a diverged σ^{70} family protein, contacting the clamp helices and mediating the sequence-specific interactions with the promoter -10-like element. In addition to the $\sigma R2$ homology domain, we find that gp55 has a HLH motif, which is critical for T4 late transcription. The structure of TAC-gp45 reveals that gp33 contacts the upstream dsDNA, which is important for gp45-dependent transcription activation. The structure verifies that gp45 is located at the upstream end of TAC-gp45, in the vicinity of gp33. The gp45 protomers, to which gp33 and gp55 bind, are assigned based on the restraints of their linkers.

Reviewer #2 (Remarks to the Author):

This paper reports the EM-obtained structures of two aspects of the T4 late gene transcription machinery: 1) the open complex of RNA polymerase with the T4 late sigma factor, gp55, and the promoter template and 2) the activated complex of RNAP with gp55, the T4 coactivator (the sigma70 Region 4 substitute gp33), the sliding clamp enhancer (gp45), and the DNA. It is truly exciting to now be able to visualize these complexes, and this is a very nice contribution to the field. The visualized complexes essentially confirm every aspect of the machinery that was worked out through the elegant biochemistry by George Kassavetis and others in the lab of Peter Geiduschek.

My main concern is the tone of the paper. Although the authors do reference the Geiduschek work, they do not really reflect how their work confirms what is already known. For example, the model in Fig.6 is exactly the same as what already appears in reviews discussing this type of transcription, all built from the previous biochemistry. Furthermore, the authors state in line 69, 'Genetic and biochemical studies have elucidated some aspects of gp45-dependent transcription activation. Nevertheless, a precise mechanistic understanding of the process remains elusive, in part, because of a lack of structural information for an intact gp45-dependent transcription activation complex.' Perhaps this is true in a very technical sense, but it certainly undervalues the work that has been done previously. In addition, because of their lack of density in the SCBM, even they are not able to add these details at this time. A novice or young reader would assume from the way the paper is written that their structural analyses are providing the basis for the model in Fig. 6.

We thank the reviewer for the constructive suggestions to improve the quality of the manuscript. The tone of the manuscript has been revised according to the reviewer's suggestion. The misleading statements have been deleted in the revised version. The biochemical studies by Peter Geiduschek and colleagues have been acknowledged, as well. We also want to mention that the model is not exactly the same as what already appears in reviews. Recycle of the RNAP-gp55-gp33-gp45 quaternary complex is proposed for the first time.

I want to be clear that I am not suggesting that this paper is not an excellent contribution. I am instead saying that the work would be even more powerful, if the authors had chosen to change

their point-of-view and acknowledge specifically and in detail what has been done, what was known, and what they either confirmed or found as new. It does not diminish this work to indicate how it follows from others.

The previous work has been described in detail in the Introduction. In addition, a paragraph describing what is confirmed or found as new has been added in the Discussion. The added information is excerpted below:

Bacteriophages use bacterial RNAP to transcribe their own genes. For decades, transcription of bacteriophage T4 late genes has served as a model system to investigate mechanisms of transcription regulation. The T4 late promoters consist of a -10-like element placed at the position corresponding to the bacterial promoter -10 element ⁸. Nevertheless, there is no sequence conservation at the position corresponding to the bacterial promoter -35 element. Peter Geiduschek and colleagues developed an *in vitro* system that leads to the current understanding of T4 late transcription ⁹. It was found that T4 encoded gp55 enabled RNAP to execute low level (basal) transcription, which was further activated by the sliding clamp gp45 and its coactivator gp33.

Gp55, a highly diverged σ^{70} family protein ¹⁰, plays an essential role in T4 late transcription. To initiate basal transcription, bacterial RNAP needs to form holoenzyme with gp55, which recognizes T4 late promoters and confers the ability to form a catalytically competent RNAP-promoter open complex (RPO-gp55) ¹¹. The RNAP binding motif of gp55 has been inferred on the basis of alanine scanning mutagenesis analyzed for RNAP binding, basal and activated transcription ¹². Initial binding of RNAP-gp55 holoenzyme to DNA is not sequence-specific, while RPO-gp55 is sequence-specific and readily detected by footprinting ^{13,14}. The acquisition of sequence specificity on promoter opening implies recognition of some feature of the transcription bubble by gp55, but this has not been demonstrated directly.

Gp33 and gp45 further activate the basal transcription activity of RNAP-gp55 holoenzyme by forming a catalytically competent transcription activation complex (TAC-gp45) ¹⁴. Pull-down experiments showed that gp33 binds to the RNAP β flap ¹⁵, which was further confirmed by the crystal structure of gp33 complexed with the RNAP β flap ¹⁶. Although gp33 does not, by itself, bind to DNA ⁹, photo-crosslinking experiments showed that gp33 was proximal to the upstream double-stranded DNA (dsDNA) in TAC-gp45 ^{14,17}. Without a precise structure of TAC-gp45, whether gp33 directly contacts the upstream dsDNA is unknown. DNase I footprinting and photo-crosslinking experiments collectively suggested that gp45 was located at the upstream end of TAC-gp45, in the vicinity of the coactivator, gp33 ¹⁴. Gp45 is a homotrimer, forming a triangle with a central hole large enough to accommodate dsDNA ¹⁸. The lateral face with the protruding C-terminus presents a hydrophobic patch on each protomer that serves as a binding site for the sliding clamp binding motif (SCBM) that is attached to the body of gp55 and gp33 through a linker. *In vitro* transcription assays with SCBM truncated proteins showed that the SCBM of gp33 is essential for transcription activation, while loss of the SCBM of gp55 impairs but does

not abolish transcription activation^{19,20}.

For the first time, we determine the precise structures of T4 late gene transcription machinery. The structures unequivocally show that a sliding clamp can turn on transcription by physically tethering RNAP to the DNA. Our structures confirm that gp55 is a diverged σ^{70} family protein, contacting the clamp helices and mediating the sequence-specific interactions with the promoter -10-like element. In addition to the $\sigma R2$ homology domain, we find that gp55 has a HLH motif, which is critical for T4 late transcription. The structure of TAC-gp45 reveals that gp33 contacts the upstream dsDNA, which is important for gp45-dependent transcription activation. The structure verifies that gp45 is located at the upstream end of TAC-gp45, in the vicinity of gp33. The gp45 protomers, to which gp33 and gp55 bind, are assigned based on the restraints of their linkers.

Other points:

1) I am confused why the authors chose to use the Mango readout to test the transcription activities of their proteins. Why didn't they observe transcription directly? The level of enhancement observed in Fig. S1C in the presence of complete system with the enhancer is low. Do the authors have an explanation for that? They should point out how their level of enhancement compares with previous work.

The import of radioactive material is compromised during the COVID-19 pandemic. Therefore, we resort to fluorescence. Although kinetic analysis indicates that transcription can be activated by ~320 fold (Kolesky et al JMB 2002), the RNA product does not increase so much at experimental RNAP concentrations and time scales. The level of enhancement observed in Mango III experiments (~3 fold) is similar to the previous work using radioactive transcription assay (Figure 9A in Kolesky et al JMB 2002, Figure 5A in Nechaev et al JMB 2008, and Figure S2 in Twist et al PNAS 2011). The comparison between the Mango III transcription assay and the radioactive transcription assay has been added in line 107.

2) Fig. S2 needs much more information. What were the other 75% of the particles with the TAC-gp45 molecule that were not refined? What was class I, which was not refined for the RPo? How did they determine which class to use?

Figure S2 has been strengthened as below:

Figure S2. Data processing pipeline.

(A) Initial 3D classification results in a reconstruction with densities for gp45 (TAC-gp45) and a reconstruction without densities for gp45 (RPo-gp55). For RPo-gp55, further 3D classification with 4 classes is performed. The density map of the upstream dsDNA of class 1 (in the dashed circle) is weak compared with the other three classes. Therefore, only classes 2-4 are combined for refinement. For TAC-gp45, masked 3D classification leads to four classes with slightly different conformations of gp45 relative to RNAP, which is consistent with the model that gp45 is tethered to RNAP through flexible linkers. Class 3

is analyzed in detail because its resolution is the highest among the four classes. Predicted local resolution of gp45 is indicated below each class.

(B) Superposition of four classes of TAC-gp45 shows slightly different conformations of gp45 relative to RNAP.

3) An input control is needed for Fig. 2E – Accounting for the smaller size of the gp55 truncated protein at 114 aa, it is not clear to me that the binding is significantly affected. Please quantify and show std deviation.

The pull-down experiments are quantified according to the reviewer's suggestion. It turns out that truncation of the HLH motif does not affect RNAP-gp55 binding. To figure out why truncation of the HLH motif compromises RPo-gp55 formation and gp55-dependent transcription, we check the structures again and find that the HLH motif chaperons the template-strand ssDNA in the active center cleft. In particular, residue R114 makes an electrostatic interaction with the DNA backbone phosphate at position -11. Therefore, we infer that the major function of the HLH motif is to stabilize the template-strand ssDNA of the transcription bubble. Consistently, alanine substitution of residue R114 compromises RPo-gp55 formation and gp55-dependent transcription.

Quantification of the pull-down experiments have been added as Figure S7B. The interaction of the HLH motif with the template-strand ssDNA has been described in line 163 and presented in Figure 4.

Reviewer #3 (Remarks to the Author):

Activation of late phage T4 transcription is an important and interesting phenomenon, as it occurs by a distinct mechanism, different from other transcription activatory mechanisms. This is a clearly presented piece of structural biology that, to my knowledge, breaks new ground, and it finishes with a convincing model.

Two issues trouble me:

1. does this mode of activation apply to other systems?
2. does the RNAP/gp33/gp55/gp45 quaternary complex really stay together during elongation?

To my mind, more discussion is needed concerning both points.

We thank the reviewer for the constructive suggestions to improve the quality of the manuscript. Other instances of transcription activation by a sliding clamp have not been found in nature. Nevertheless, fusion proteins linking the transcription activation domain of herpes simplex virus VP16 protein to a sliding clamp have been shown to activate transcription by yeast and *Drosophila* RNA polymerase II *in vitro* (Ouhammouch et al PNAS 1997).

If the quaternary complex stays together during elongation, gp33 would decrease the accessibility of the FTH to the general transcription factor NusA, which plays key roles in p-

dependent transcription termination and transcription-translation coupling. Therefore, both processes would be compromised during T4 late transcription. Although our structures are consistent with the model that the quaternary complex stays together during elongation, we cannot rule out the possibility that gp33, gp55, and gp45 might fall off RNAP during elongation.

The above discussion has been added in lines 255 and 267.